# Study on Quality Control of Compound *Anoectochilus roxburghii* (Wall.) Lindl. by Liquid Chromatography–Tandem Mass Spectrometry

**DOI:** 10.3390/molecules27134130

**Published:** 2022-06-27

**Authors:** Qiuhua Zhang, Lingyi Huang, Youjia Wu, Liying Huang, Xiaowen Xu, Renyi Lin

**Affiliations:** School of Pharmacy, Fujian Medical University, Fuzhou 350122, China; autumnhua9@163.com (Q.Z.); lhuang46@sina.com (L.H.); yoga0411@hotmail.com (Y.W.); xxw184500@163.com (X.X.); fjmulry@163.com (R.L.)

**Keywords:** LC–MS/MS, *Anoectochilus roxburghii* (Wall.) Lindl. and its compound oral liquid, *Ganoderma lucidum*, analysis, quality control

## Abstract

Compound *Anoectochilus roxburghii* (Wall.) Lindl. (*A. roxburghii*) oral liquid (CAROL) is a hospital preparation of *A. roxburghii* and *Ganoderma lucidum* (*G. lucidum*)*,* which have hepatoprotective effects. Eight active components (five nucleosides/nucleobases and three triterpenoid acids) in CAROL, *A. roxburghii*, and *G. lucidum* were simultaneously detected by high-performance liquid chromatography–tandem mass spectrometry (LC–MS/MS). The multiple reaction monitoring (MRM) mode was applied for the detection of analytes. These eight compounds were separated well within 12 min and quantified using the internal standard working curve method. The method showed good linearity (*R*^2^ > 0.9935) and high sensitivity (limit of detection = 0.29 ng/mL). The analyte recovery ranged from 85.07% to 97.50% (relative standard deviation < 3.31%). The content of the target analytes in four batches of CAROL, and the raw materials of *G. lucidum* and *A. roxburghii* from the five regions was determined using this method. The contents of guanosine and ganoderic acid A in four batches of oral liquid were high and stabilized and could be recommended as quality markers (Q-marker) for CAROL. Simultaneous qualitative and quantitative analysis of nucleosides and triterpenoid acids in CAROL, *A. roxburghii*, and *G. lucidum* by LC–MS/MS based on the MRM model was reported for the first time. The proposed method provides a sensitive, rapid, and reliable approach for the quality control of Chinese medicinal products.

## 1. Introduction

Compound *Anoectochilus roxburghii* (Wall.) Lindl. oral liquid (CAROL) is a traditional Chinese medicinal formula approved by a hospital in Fujian as its hospital preparation; they found that CAROL has heat-clearing, detoxifying, and blood-cooling effects. It is considered as an effective treatment for chronic viral hepatitis B [1]. The formula is prepared from two precious Traditional Chinese Medicines (TCMs), namely, *Anoectochilus roxburghii* (Wall.) Lindl. (*A. roxburghii*, Jin Xian Lian) and *Ganoderma lucidum* (Ling Zhi) [2]. *A. roxburghii* enjoys the reputation of being the “King of Medicine” and “magic medicine”; it has curative effects on carbon tetrachloride (CCl_4_)-induced liver damage, alcoholic liver disease, non-alcoholic fatty liver disease, autoimmune hepatitis, and chronic hepatitis B [3,4]. *G. lucidum* is called “fairy grass” and has a medicinal history of more than 2000 years [5]. *G. lucidum* can enhance anti-inflammatory activity of the body and effectively treat chronic liver disease [6,7]. Nucleoside compounds are the basic constituents of living organisms, and they are found in both *A. roxburghii* and *G. lucidum*. They play an important role in protecting the liver and resisting viruses [8,9]. Nucleosides such as adenine, uridine, and guanosine have shown anti-inflammatory, anti-oxidant, and liver metabolism-regulating functions [10,11,12]. Deoxynucleosides have attracted people’s attention because of their antiviral and antitumor activities [13]. Triterpenoids are among the bioactive constituents of *G. lucidum* with anti-liver injury effects [14,15]. Ganoderic acid A and F are representative ingredients from *G. lucidum*, having antinociceptive, antioxidative, anticancer, and hepatoprotective activities [16,17]. Lucidenic acid A has shown anti-invasion activity against human hepatoma carcinoma cells [18].

Therefore, it is important to develop a more sensitive and efficient analytical method for the determination of more bioactive components in CAROL for its quality assurance. It also provides a material basis for CAROL’s hepatoprotective pharmacological activity.

Although quantitative analysis of single or a few compounds in *A. roxburghii* or *G. lucidum* has been reported by high-performance liquid chromatography (HPLC) [19], liquid chromatography–mass spectrometry (LC–MS) [20], micellar electrokinetic chromatography (MEKC) [21], and indirect competitive enzyme-linked immunosorbent assay (ELISA) [22], the simultaneous qualitative and quantitative analysis of nucleosides and triterpenoid acids in CAROL has not been conducted by liquid chromatography–tandem mass spectrometry (LC–MS/MS) based on the multiple reaction monitoring (MRM) mode.

Recently, LC–MS/MS has been widely used in the analysis of complex Chinese herbal medicines and their preparations [23,24]. In addition, the MRM mode can be used to selectively detect and quantify targets based on the screening of specified molecular ion-to-product ion transitions [25]. This mode can provide high sensitivity and low detection and quantitation limits, and it has been proposed as an alternative approach for rapid analysis [26]. Moreover, the LC–MS/MS method based on the MRM model is especially suitable for the analysis of active components in compound oral liquid with a complex matrix, because it does not require the complete separation of components.

In the present paper, an LC–MS/MS method within 12 min for the simultaneous determination of eight bioactive compounds in CAROL, *A. roxburghii*, and *G. lucidum* was established for the first time. The eight target analytes are adenine, uridine, 2′-deoxyuridine, 2′-deoxyadenosine, guanosine, lucidenic acid A, ganoderic acid F, and ganoderic acid A. Their chemical structures are shown in Figure 1. MRM was applied to detect target compounds by the internal standard working curve method. This method was validated and successfully applied for the analysis of four batches of CAROL, a water extract of *A. roxburghii*, a water extract of *G. lucidum*, and an alcohol extract of *G. lucidum*. Tenofovir was used as an internal standard (IS) to ensure the accuracy of quantitative measurement. The content of the target analytes in the raw materials of *G. lucidum* and *A. roxburghii* from the five regions was determined. Simultaneous analysis of nucleosides, nucleobase, and triterpenoid acid in CAROL by LC–MS/MS based on the MRM model was reported for the first time. The proposed method provides a rapid and reliable approach for the quality control of Chinese medicine products.

## 2. Results and Discussion

### 2.1. Optimization of Chromatographic and MS Conditions

Favorable chromatographic conditions with good peak shapes and resolutions in a short time were obtained by comparing different columns and mobile phase systems. We compared several reverse phase columns, such as the Acquity BEH C18, Ultimate^®^ XB-C18, and Poroshell 120 SB-C18 columns. The results indicated that the Poroshell 120 SB-C18 column was more suitable for the separation of all tested compounds. The Poroshell 120 SB-C18 column was selected for the experiment because it has good column efficiency and can complete the separation of the analytes in a short time and maintain a good peak shape with a low column pressure. Considering that the polarity of nucleotides and nucleosides is relatively strong [27], MeOH–water was selected as the mobile phase because of the weaker elution capacity of MeOH than that of ACN, which could effectively separate these highly polar compounds. Furthermore, the addition of formic acid into the aqueous phase could enhance the MS signal of nucleotides and ganoderic acid, which was not observed by adding ammonium acetate or ammonium formate. Water containing 0.1%, 0.2%, and 0.3% formic acid was tested further. The results demonstrated that water with 0.2% formic acid could provide the best peak shape for eight analyses (figure not shown). Therefore, water with 0.2% formic acid (A)–MeOH (B) was selected as the mobile phase for separating these components. All compounds were tested in the full-scan mode in both positive and negative channels to optimize the MS conditions. Ganoderic acid A and lucidenic acid A showed high sensitivity and clearer mass spectra in the negative mode, while the other substances obtained good mass spectra peak in the positive mode. Among them, tenofovir is a nucleoside antiviral drug that has a similar structure to nucleosides and does not exist in natural medicinal materials, so it was used as an internal standard (IS). The response signals of tenofovir were detected by scanning in positive and negative ion modes. Therefore, the polarity switching modes of QqQ MS were used to achieve the highest response intensities of various types of components. The collision energy (CE) was also optimized to obtain the best mass response according to the kurtosis ratio of MRM and the mass response to obtain the optimal collision energy (CE). For example, adenine produced various fragment ions under different collision energies between 10 eV and 50 eV, and the ion intensities were different. We selected the highest response signal of 136.1 > 119.1 (CE: −16 eV) as the quantitative transition. All results are summarized in Table 1.

### 2.2. Method Validation

The method was validated to determine the linear range, limit of detection (LOD), limit of quantification (LOQ), precision, repeatability, stability, and recovery. The results are shown in Table 2.

#### 2.2.1. Linear Range, LOQ, and LOD

The linearity of each point on the calibration curve was determined by plotting the analyte/IS peak area ratio (*Y*) against the concentrations of analytes (*X*). The standard curves of all components showed good linearity within a specific linear range (*R*^2^ > 0.9935).

The LODs and LOQs of each compound were determined at signal-to-noise (S/N) ratios of 3 and 10, respectively. The LOD and LOQ for all standard analytes were in the range of 0.29–70.4 and 0.953–235 ng/mL, respectively, indicating that this method was sensitive for quantitative measurement.

#### 2.2.2. Precision, Repeatability, and Stability

Precision was certified by evaluating the relative standard deviations (RSDs) of compounds’ peak areas of six different CAROL samples from the same batch. The repeatability was determined by the RSD of six replicates of one CAROL working solution. The RSD (*n* = 6) values of all tested compounds’ precision were less than 0.58%, and the values of repeatability were less than 1.97%, indicating that the method was feasible and reproducible.

To evaluate the stability of the proposed method, the same CAROL sample was injected at static times (0, 1, 2, 4, 6, 8, 12, 24, 48, and 72 h) at room temperature, and their peak areas were used for calculation. The RSD values of the peak areas of all analytes were less than 4.01%, indicating that the CAROL was stable up to this time.

#### 2.2.3. Recovery

The recovery rate of eight analytes in CAROL was performed as follows: take 0.5 mL of CAROL; accurately add an appropriate amount of composite stock solution according to the three levels of low, medium, and high; add 1 mL of internal standard solution; and make up the volume to 5 mL. The resulting concentrations were 80%, 100%, and 120% of the actual sample concentrations, respectively. The recoveries of *A. roxburghii* (0.5 g) and *G. lucidum* (1.0 g) were determined by adding the standard substance with a similar concentration to the actual sample to the herb samples and then extracting and analyzing them as described in Section 3.4. The average recovery rate was calculated using the following formula: recovery (%) = (total amount after spiking – original amount in the sample)/spiked amount × 100%. The recoveries of the eight analytes in CAROL were determined, and the results are shown in the Appendix A. The recoveries of components in the CAROL solution were between 85.07% and 97.50%, and the RSD values were less than 3.31%. In addition, the recoveries of nucleosides and ganoderic acids in all extracts were between 81.86% and 97.61%, and the RSD values were less than 1.10% (Appendix A).

### 2.3. Sample Analysis

The established method was applied to analyze the contents of eight compounds in actual samples. MRM chromatograms of compounds are shown in Figure 2A–C. As shown in Figure 2A–C, the differences in retention times of the five nucleosides/nucleobases were small, and the retention times of the three triterpenoid acids were also very similar, but quantitative analysis could still be performed. This further proved that the LC–MS/MS method based on the MRM model is especially suitable for the analysis of active components in compound oral liquid and TCMs with a complex matrix because it does not require the complete separation of components.

The proposed method was used to determine the amounts of compounds in four different batches of CAROL, *A. roxburghii*, and *G. lucidum* (collected in five different regions). Representative chromatograms of working solutions and samples are shown in Figure 3.

The eight marker compounds were identified unambiguously by comparing the retention times and product ions in the MRM mode of reference standards, and the dissociation rules of precursor ions were consistent with the literature [28,29,30].

#### 2.3.1. Analysis of CAROL

This method was used for the detection of all the tested compounds from four batches of CAROL samples. The chromatographic separation is illustrated in Figure 3. Since the sample was tested by five times dilution of CAROL, the actual sample concentration was obtained by multiplying the measurement result by five, and the results are shown in Table 3. In this experiment, ganoderic acid A exhibited the highest content with an average of 46.35 μg/mL. The average concentrations of lucidenic acid A and ganoderic acid F were 2.855 and 40.13 μg/mL, respectively. Among the five tested nucleosides, guanosine showed the highest content average of 14.79 μg/mL, and 2′-deoxyadenosine exhibited the minimum content average of 0.015 μg/mL. The detailed data are summarized in Table 3. In comparison with the experimental results of Xu et al. [19], the LOQ values of the co-measured substances of ganoderic acid A, uridine, and guanosine were much lower. This finding was obtained because LC–MS/MS techniques provided better peak separation, mass measurement accuracy, and high accuracy/resolution compared with conventional LC [31]. The SPSS25.0 software was used to statistically analyze the concentration of compounds in different batches of CAROL, and one-way analysis of variance (ANOVA) was performed on the eight tested components in four batches of CAROL at the 0.05 level. Except for guanosine and ganoderic acid A, the other tested components showed *p* values below 0.05. The *p*-values for guanosine and ganoderic acid A were 0.560 and 0.245, respectively, while the *p*-values for the others were all below 0.05, indicating extremely significant differences among the six substances in the four batches of CAROL. The tested compounds have made great contributions to the liver-protecting effects of CAROL. However, the significant differences among different batches indicated that the CAROL preparation was not stable enough, causing possible concern about the efficacy and safety of CAROL. That is why quality control of CAROL is necessary.

The results showed that compared with the other six analytes, the contents of guanosine and ganoderic acid A in CAROL were higher, and there was no significant difference in content between different batches. According to the literature, guanosine and ganoderic acid A have good anti-inflammatory and hepatoprotective activities [32,33,34]. Combined with the definition of the Q-marker [35], these two substances (two types of active compounds) were recommended as Q-markers for the quality control of the oral liquid.

#### 2.3.2. Analysis of Extracts of *A. Roxburghii* and *G. lucidum*

The chromatographic separation of the tested constituents from the extracts with water of *A. roxburghii* and *G. lucidum* is illustrated in Figure 2, and the extract concentrations are summarized in Table 4. As shown in the table, the contents of nucleosides in *A. roxburghii* were 1.137–448.4 μg/g, while those in *G. lucidum* were 0.4449–255.3 μg/g. More nucleosides were present in *A. roxburghii* than in *G. lucidum*. In addition, the total amount of nucleosides in *A. roxburghii* from Mingxi region reached 619.9 μg/g, which was the highest among the five detection areas, while the total amounts in other areas were only 519.6, 532.7, 547.8, and 582.9 μg/g. Moreover, the total amount of nucleosides in *G. lucidum* from Sanming region reached 452.2 μg/g, while the total amounts in other areas were 114.5, 143.8, 153.8, and 185.1 μg/g. These results indicated differences in the content of the same medicinal materials from different places. This finding can be attributed to environmental factors such as temperature, soil humidity, and salinity, which have important effects on plant growth [36]. These factors also affect the accumulation of active ingredients in *A. roxburghii* and *G. lucidum*.

The raw materials of TCMs are the source of the quality differences of Chinese patent medicines, and the production process greatly influences the quality consistency [37]. Therefore, the stability and uniformity of CAROL can be improved by maintaining the same quality of Chinese medicinal materials selected in each batch.

Only extraction reagents that differ were compared in the experiment. The contents of ganoderic acid in water and ethanol extracts of *G. lucidum* are shown in Table 4. The results show that the effect of ethanol extraction of ganoderic acid was obviously better than that of water extraction. Ganoderic acid A and lucidenic acid A were not detected in the water extracts of *G. lucidum* from Sanming, Longyan, and Zhangzhou, but they were detected in the ethanol extracts. The total amounts of ganoderic acid in the water extracts from Ningde and Nanping were 556.5 and 764.31 μg/g, respectively, but the numbers could reach 1002.65 and 1164.88 μg/g, respectively, in ethanol extraction. Therefore, the content of ganoderic acid could be effectively increased by adding the proper amount of ethanol during CAROL preparation.

In addition, when analyzing the *G. lucidum* from Zhangzhou, lucidenic acid A and ganoderic acid A were not detected in the water extracts. Moreover, the contents in the ethanol extracts were below the LOQs. Differences in the origin of *G. lucidum* or improper storage may cause the medicinal materials to deteriorate.

## 3. Materials and Methods

### 3.1. Chemicals, Materials, and Reagents

The standards of adenine, uridine, 2′-deoxyuridine, guanosine, and tenofovir were obtained from Shanghai Aladdin Bio-Chem Technology Co., Ltd. (Shanghai, China). 2′-Deoxyadenosine was obtained from Shanghai Macklin Bio-Chem Technology Co., Ltd. (Shanghai, China). Ganoderic acid A and F were obtained from Shanghai Yuanye Bio-Technology Co., Ltd. (Shanghai, China). Lucidenic acid A was obtained from Shanghai Tansoole Bio-Technology Co., Ltd. (Shanghai, China). The purities of all the above ingredients were higher than 98%. Chromatographic-grade methanol (MeOH) and acetonitrile (ACN) were obtained from Sigma Aldrich Reagent Co., Ltd. (St., Louis, MO, USA). The analytical-grade absolute ethanol used for extraction was obtained from Shanghai Tansoole Bio-Technology Co., Ltd. (Shanghai, China). HPLC-grade formic acid was obtained from Shanghai Aladdin Bio-Chem Technology Co., Ltd. (Shanghai, China). Double distilled water was used in this study.

CAROL was obtained from the Mengchao Hepatobiliary Hospital of Fujian Medical University (Fujian Medicine Z04107018, Fuzhou, China). Five batches of raw *G. lucidum* and *A. roxburghii* samples were collected from different regions. *G. lucidum* was collected from Longyan, Sanming, Zhangzhou, Ningde, and Nanping; *A. roxburghii* was obtained from Mingxi, Nanjing, Yong’an, Fuqing, and Taiwan.

### 3.2. Instrumentation

A triple quadrupole mass spectrometer 8040 (Shimadzu Corporation, Kyoto, Japan), a UV-2450 spectrometer (Shimadzu Corporation, Kyoto, Japan), a KQ-100TDV ultrasonic water bath with temperature control (Kunshan Ultrasonic Instrument Co., Ltd., Kunshan China), an AR224CN electronic balance (Ohaus Instrument Co., Ltd., Changzhou, China), a TG16-WS high speed bench centrifuge (Hunan Michael Experimental Instrument Co., Ltd., Hunan, China), and an SZ-93 automatic double-pure water distillatory (Shanghai YaRong Biochemical Instrument Factory, Shanghai, China) were used.

### 3.3. Preparation of Standard Solutions

The reference standards of adenine, uridine, 2′-deoxyuridine, and 2′-deoxyadenosine, and the internal standard tenofovir were accurately weighed and dissolved in water, and guanosine, ganoderic acid A, lucidenic acid A, and ganoderic acid F were dissolved in MeOH. The concentration of each standard stock solution was 1.000 mg/mL, and they were stored at 4 °C. The composite stock solution was prepared by mixing the eight analytes in appropriate proportions. The composite stock solution was diluted with water to prepare a series of different concentrations, and the same amount of internal standard solution was added to obtain the working solution. The concentration of the internal standard substance in each working solution was 5.00 μg/mL. These working solutions were stored at 4 °C and filtered through a 0.22-μm syringe filters before injection into the LC–MS/MS system.

### 3.4. Sample Preparation

#### 3.4.1. Working Solution of CAROL

CAROL was centrifuged at 12,000 rpm for 2 min. Then, 1 mL of the supernatant was accurately aspirated, added with 1 mL of internal standard solution, and then diluted with water to a final volume of 5 mL. Finally, these solutions were filtered through syringe filters (0.22 μm) before injection into the LC–MS/MS system.

#### 3.4.2. Extract of Nucleosides and Nucleobases

The *A. roxburghii* powder was filtered through a 200-mesh sieve and dried in a vacuum drying oven at 60 °C for 6 h. The dried powder was accurately weighed (0.500 g ± 0.005 g) and subjected to ultra-sonication with 20 mL of water. The extraction was conducted at 50 °C for 50 min, followed by centrifugation at 6500 rpm for 15 min. Then, the mixture was filtered, and the residue was extracted again by using the same approach. The solution was collected in a 50 mL volumetric flask and made up to the mark with water.

*G. lucidum* powder was sieved through 15 mesh and dried in a vacuum drying oven at 60 °C for 6 h. Other operations were conducted using the same extraction process for *A. roxburghii*.

#### 3.4.3. Extract of Ganoderic Acid

For the water extract, five batches of *G. lucidum* powder were sieved through a 15 mesh and dried in a vacuum drying oven at 60 °C for 6 h. Different samples were accurately weighed (1.000 g ± 0.001 g) and mixed with 20 mL of water. Then these solutions were sonicated for 60 min. After centrifugation at 6500 rpm for 15 min, the mixture was filtered, and the residue was extracted again by using the same approach. The solution was collected in a 50 mL volumetric flask and made up to the mark with water.

For the ethanol extract, the extracted solution was obtained following the same operation as the water extraction method. Then the absolute ethanol was evaporated in a vacuum drying oven at 60 °C for 24 h. Afterward, the dry solid was dissolved in 1 mL of methanol, carefully transferred into a 50 mL volumetric flask, and made up to the mark with water.

All extracts were stored in a refrigerator at 4 °C before use. Further, these solutions were added with the internal standard, diluted with water to the proper concentrations, and then filtered through syringe filters (0.22 μm) before injection into the LC–MS/MS system.

### 3.5. Instrument Conditions

The samples were analyzed using an LC–MS/MS 8040 system equipped with an LC-20AD binary pump, a CTO-20A column oven, a SIL-20AC autosampler, and an FCV-20A controller. An Agilent Poroshell 120 SB-C18 column (3.0 × 100 mm, 2.7 μm) with a Poroshell SB C18 guard column (3 × 5 mm, 2.7 μm) was applied for the chromatographic separation. The flow rate was kept constant at 0.3 mL/min. The column temperature was maintained at 40 °C. The mobile phase consisted of water containing 0.2% (*v*/*v*) formic acid (A) and MeOH (B). The elution program was as follows: 0–5 min, 5% B; 5–5.1 min, 5–80% B; 5.1–15 min, 80% B. The system was flushed with 90% methanol, and the samples were injected after equilibrating with 5% B. The injection volume was 10 μL.

Mass spectrometry was conducted using a Shimadzu LC–MS/MS 8040 triple quadrupole mass spectrometer equipped with an electrospray ionization (ESI) interface. MRM was utilized for the detection of analytes in positive or negative mode. The conditions of the ESI source were as follows: heat block temperature was 400 °C; DL temperature was 250 °C; nebulizing gas (N_2_) flow was 3 L/min; and drying gas flow (N_2_) was 15 L/min.

### 3.6. Sample Determination

All samples were prepared according to Section 2.3. Then, the compounds were determined by the retention time and MRM transition of each compound. In this study, the response signals of tenofovir were detected by scanning in positive and negative ion mode. According to the signal of the eight analytes scanned in positive and negative ion modes, the one with the large signal value was selected as the quantitative scanning mode. Ganoderic acid A and lucidenic acid A showed high sensitivity and clearer mass spectra in the negative ion mode, while the other substances obtained good mass spectra peaks in the positive ion mode. Positive or negative ion mode was selected for the internal standard according to the scanning mode of the analyte. The concentrations of the eight selected analytes were calculated based on the respective internal standard calibration curves.

## 4. Conclusions

In this study, a sensitive, rapid, and accurate method for the simultaneous determination of the eight bioactive components in CAROL, *A. roxburghii*, and *G. lucidum* was established and validated.

Simultaneous analysis of nucleosides, nucleobase, and triterpenoid acid in CAROL by LC–MS/MS based on the MRM model was reported for the first time. The method was applied to determine the content of the target analytes from the extracts of *A. roxburghii* and *G. lucidum* extracts. To the best of our knowledge, 2’-deoxyuridine in *A. roxburghii* was quantitatively analyzed for the first time. Using guanosine and ganoderic acid A as the Q-markers of CAROL will be beneficial for the quality control of the oral liquid and, further, to ensure the stability and efficacy of the product. This work can provide useful information for the quality control of CAROL and related medicinal products.

## Figures and Tables

**Figure 1 molecules-27-04130-f001:**
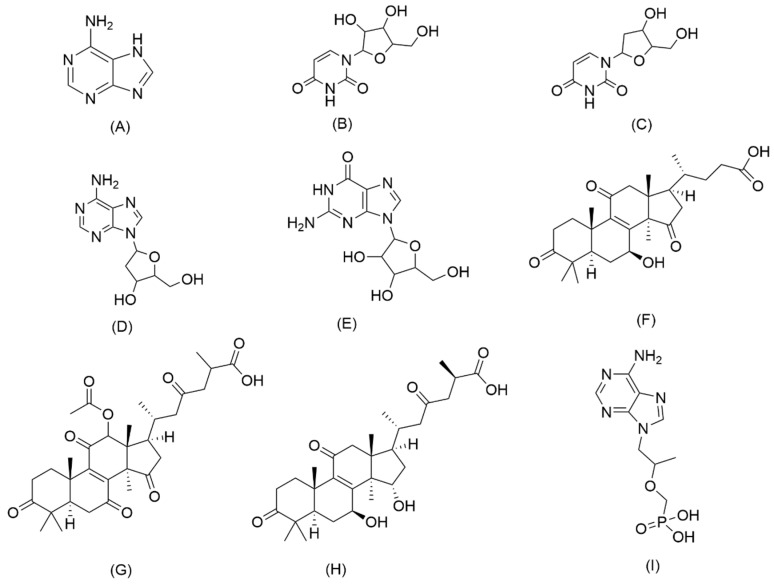
Chemical structures of eight analytes and the internal standard: (**A**) adenine; (**B**) uridine; (**C**) 2’-deoxyuridine; (**D**) 2’-deoxyadenosine; (**E**) guanosine; (**F**) lucidenic acid A; (**G**) ganoderic acid F; (**H**) ganoderic acid A; (**I**) tenofovir (Internal standard, IS).

**Figure 2 molecules-27-04130-f002:**
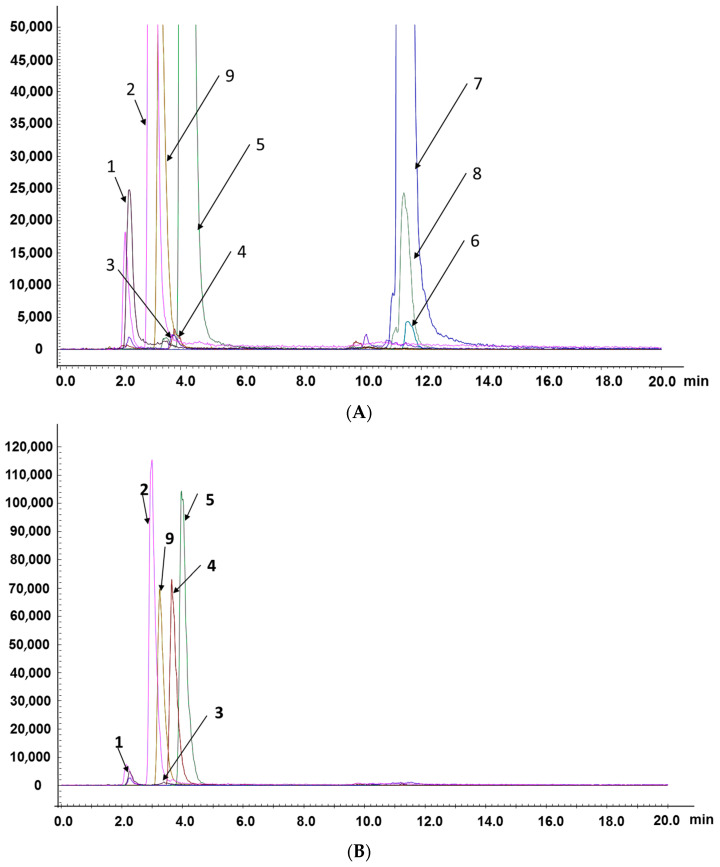
Representative MRM chromatograms of compounds: 1—adenine; 2—uridine; 3—2’-deoxyuridine; 4—2’-deoxyadenosine; 5—guanosine; 6—lucidenic acid A; 7—ganoderic acid F; 8—ganoderic acid A; 9—tenofovir (internal standard). (**A**) CAROL; (**B**) water extract of *A. roxburghii*; (**C**) ethanol extract of *G. lucidum*.

**Figure 3 molecules-27-04130-f003:**
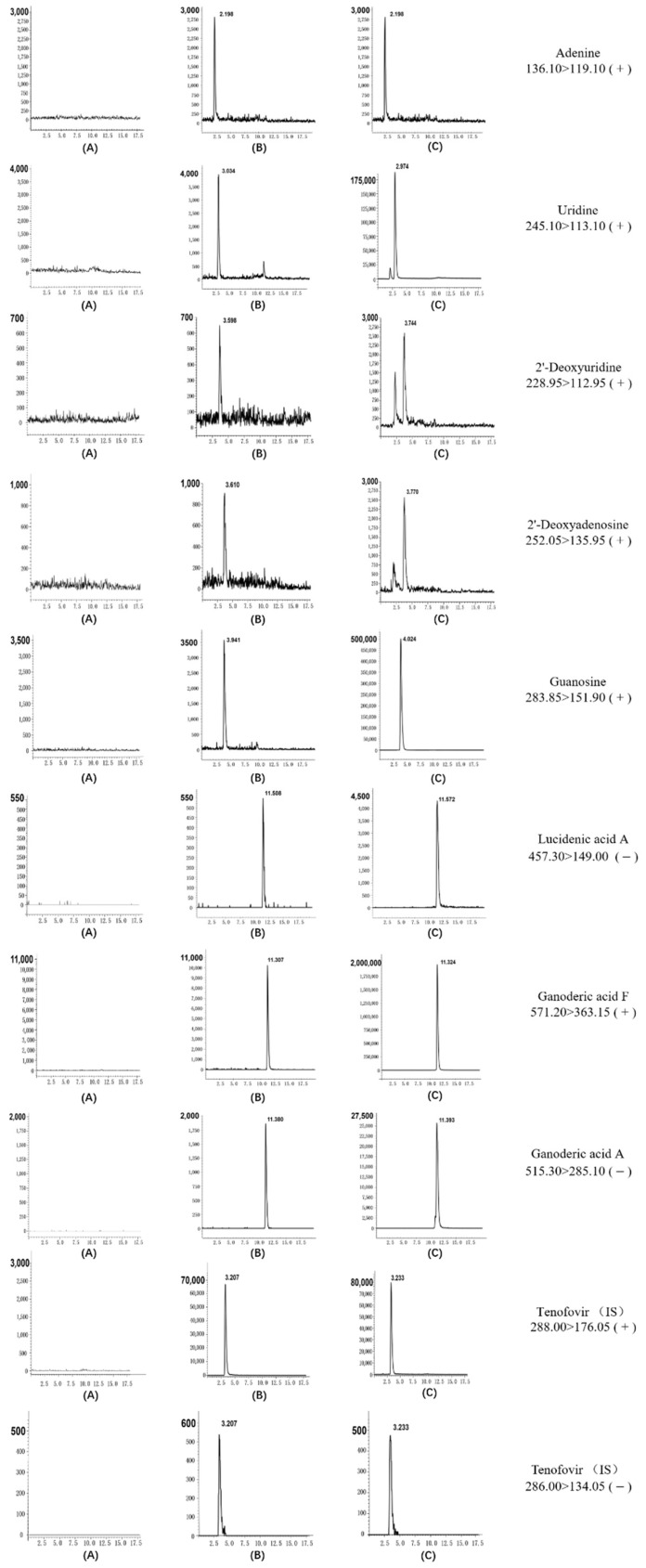
Representative extract ion chromatograms of eight analytes and internal standards: (**A**) blank; (**B**) standard substance of LOQ; (**C**) CAROL.

**Table 1 molecules-27-04130-t001:** MS parameters of eight analytes and IS.

Analytes	MW	Rt (min)	Polarity	Q1 Mass (*m*/*z*)	Q3 Mass (*m*/*z*)	CE (eV)
Adenine	135.127	2.203	(+)	136.1 [M + H]^+^	119.1 [M + H − NH_3_]^+^ *	−16
					64.9 [M + H − C_4_H_4_N − NH_3_]^+^	−41
Uridine	244.201	2.967	(+)	245.1 [M + H]^+^	113.1 [M + H − C_5_H_8_O_4_]^+^ *	−16
					73.0 [M + H − C_5_H_8_O_4_ − NH_3_ − 2H_2_O]^+^	−24
2’-Deoxyuridine	228.202	3.662	(+)	228.9 [M + H]^+^	112.9 [M + H − C_5_H_8_O_3_]^+^ *	−10
					73.0 [M + H − C_5_H_8_O_4_ − NH_3_ − 2H_2_O]^+^	−20
2’-Deoxyadenosine	251.242	3.648	(+)	252.1 [M + H]^+^	135.9 [M + H − C_5_H_8_O_3_]^+^ *	−16
					119.1 [M + H − C_5_H_8_O_3_ − NH_3_]^+^	−41
Guanosine	283.241	3.985	(+)	283.9 [M + H]^+^	151.9 [M + H − C_5_H_8_O_3_]^+^ *	−12
					134.9 [M + H − C_5_H_8_O_3_ − NH_3_]^+^	−37
Lucidenic acid A	458.587	11.521	(−)	457.3 [M-H]^−^	149.0 [M − H − C_18_H_24_O_4_]^−^ *	44
					287.0 [M − H − C_6_H_2_O_6_]^−^	35
Ganoderic acid F	570.670	11.363	(+)	571.2 [M + H]^+^	363.1 [M + H − C_11_H_14_O_4_]^+^ *	−21
					225.0 [M + H − C1_7_H_28_O_8_]^+^	−39
Ganoderic acid A	516.670	11.306	(−)	515.3 [M − H]^−^	285.1 [M − H − C_12_H_20_O_4_]^−^ *	45
					300.15 [M − H − C_13_H_23_O_4_]^−^	33
Tenofovir (IS)	287.22	3.233	(+)	288.0 [M + H]^+^	176.0 [M + H − CH_4_NO_3_P]^+^ *	−25
			(−)	286.0 [M − H]^−^	134.0 [M + H − C_4_H_9_O_3_P]^−^ *	30

* Quantification transition.

**Table 2 molecules-27-04130-t002:** The linearity, LOD, LOQ, precision, repeatability, and stability of the method.

Compounds	Calibration Curves	*R* ^2^	Linear Range (ng/mL)	Precision (*n* = 6, RSD, %)	Repeatability (*n* = 6, RSD, %)	Stability (*n* = 6, RSD, %)	LOD (ng/mL)	LOQ (ng/mL)
Adenine	Y = 0.7672X + 0.02162	0.9953	15.62–1000	0.31	0.92	1.84	4.69	15.6
Uridine	Y = 1.0454X + 0.01534	0.9953	31.25–4000	0.58	1.97	4.01	9.38	31.2
2’-Deoxyuridine	Y = 0.1556X + 0.002708	0.9965	3.906–500	0.33	0.67	1.53	1.18	3.91
2’-Deoxyadenosine	Y = 7.5340X + 0.005171	0.994	0.9531–250	0.14	1.35	1.57	0.29	0.953
Guanosine	Y = 2.4068X + 0.01341	0.9961	15.62–4000	0.26	1.80	1.23	4.69	15.6
Lucidenic acid A	Y = 1.3742X + 0.02929	0.9964	31.25–2000	0.49	1.53	1.55	9.38	31.3
Ganoderic acid F	Y = 2.6908X + 0.0007365	0.9983	46.88–12,000	0.41	1.81	2.84	14.1	46.9
Ganoderic acid A	Y = 0.5558X + 0.1136	0.9935	234.4–15,000	0.12	1.78	2.71	70.4	235

**Table 3 molecules-27-04130-t003:** Contents of eight analytes (ng/mL) in four batches of CAROL (*n* = 3).

	Batch No.	20210201	20210203	20210204	20210502	*p*
Compounds						
Adenine		1626.54 ± 2.18	1595.54 ± 3.14	1653.64 ± 2.44	2029.50 ± 3.52	<0.05
Uridine		11,127.72 ± 13.47	11,209.10 ± 17.34	11,412.36 ± 11.93	11,732.93 ± 34.90	<0.05
2’-Deoxyuridine		122.35 ± 0.36	121.83 ± 0.16	129.06 ± 0.17	153.09 ± 0.29	<0.05
2’-Deoxyadenosine		15.58 ± 0.060	10.23 ± 0.028	5.10 ± 0.015	29.40 ± 0.11	<0.05
Guanosine		14,969.06 ± 55.49	14,748.64 ± 45.79	14,730.47 ± 6.71	14,820.49 ± 20.28	0.560
Lucidenic acid A		2780.99 ± 4.44	2913.85 ± 8.91	2947.70 ± 5.56	2776.88 ± 2.11	<0.05
Ganoderic acid F		37,112.37 ± 8.54	36,957.62 ± 133.43	39,059.32 ± 87.57	47,380.18 ± 24.74	<0.05
Ganoderic acid A		46,319.64 ± 10.92	46,224.43 ± 40.81	46,314.44 ± 14.88	46,561.25 ± 106.33	0.245

**Table 4 molecules-27-04130-t004:** Contents of the analytes in the extracts of *A. roxburghii* and *G. lucidum*.

**Analytes**	**Content of *A. roxburghii* (μg/g)**					**Content of *G. lucidum* (μg/g)**				
	**Yongan**	**Taiwan**	**Nanjing**	**Mingxi**	**Fuqing**	**Longyan**	**Sanming**	**Zhangzhou**	**Ningde**	**Nanping**
Adenine	102.2	9.753	7.917	4.574	32.84	1.635	1.795	2.189	7.207	13.63
Uridine	319.7	359.5	372.4	448.4	213.4	122.9	255.3	102.2	53.43	68.69
2’-Deoxyuridine	1.787	1.247	1.410	1.137	1.138	0.4873	0.4449	N.D.	2.553	4.076
2’-Deoxyadenosine	51.27	30.96	61.78	51.01	89.66	1.134	5.220	N.D.	9.474	17.38
Guanosine	44.68	146.3	139.4	114.8	195.7	59.03	189.7	39.473	41.91	50.12
**Analytes**	**Water extract of *G. lucidum* (μg/g)**					**Ethanol extract of *G. lucidum* (μg/g)**				
	**Longyan**	**Sanming**	**Zhangzhou**	**Ningde**	**Nanping**	**Longyan**	**Sanming**	**Zhangzhou**	**Ningde**	**Nanping**
Lucidenic acid A	N.D.	N.D.	N.D.	50.81	23.78	0.7594	1.18	N.D.	84.05	35.80
Ganoderic acid F	0.5310	0.7955	0.4032	114.1	228.5	7.61	6.85	0.6047	293.1	378.8
Ganoderic acid A	N.D.	N.D.	N.D.	391.3	512.0	15.72	10.20	N.D.	625.5	750.3

N.D.: Not detected or below the LOQs. *p* of the ANOVA results of different areas are all less than 0.01.

## Data Availability

Not applicable.

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
