# Peer review of "Study on Quality Control of Compound *Anoectochilus roxburghii* (Wall.) Lindl. by Liquid Chromatography–Tandem Mass Spectrometry"

_molecules, 2022, doi:10.3390/molecules27134130_

Round 1

Reviewer 1 Report

The Authors have sufficiently answered the Reviewer's questions and included necessary corrections. In the Reviewer's opinion, the manuscript improved and can be processed further.

Reviewer 2 Report

The reviewer thinks that the authors well revised the manuscript for all issues and the manuscript is now acceptable for publication on Molecules.

This manuscript is a resubmission of an earlier submission. The following is a list of the peer review reports and author responses from that submission.

Round 1

Reviewer 1 Report

The authors challenged themselves to develop a method and apply it to quality control of several batches of traditional Chinese medicine - CAROL. Because CAROL is compounded from two herbs, they also analyzed the content of active ingredients in herbs from different sources. The results offer an approach for routine analysis and quality assurance of CAROL.

Below please find some comments that may help the authors to improve their manuscript:

  1. The list of authors appears to be incomplete (the last name is missing).
  2. Introduction:
    1. What do the authors mean by "heat-clearing" and "blood-cooling"? These statements should be supported by the outcomes of a clinical study or some research. Is this medicine used in a hospital setting without the Ethical Committee's consent?
    2. What is the origin of the herbs used to compound CAROL in the hospital? Are they from any specific region? It could be helpful to see if the contents measured in CAROL match the ones from the extracts in Table 4.
  3. Method:
    1. What other columns were tested in the method development process?
    2. What do the authors mean by "ultra-low column pressure"? What value do they have in mind?
    3. How many concentration levels were used to establish the method's precision and accuracy?
  4. Results:
    1. It would be helpful to present the p values from ANOVA in the tables that compare the active ingredient contents in CAROL.
    2. Were the analyte content statistically compared between the different A.roxburghii and G.lucidum extracts compared with ANOVA? It could be included and presented in Table 4.
    3. Why do the authors propose Ganoderic acid A and guanosine as Q-markers? Why do other studied compounds not meet the definition of the Q-marker? If not, then why they were even analyzed? Moreover, Ganoderic acid A was not found in one of the G.lucidum ethanol extracts (Zhangzhou). If not, then is this compound a good candidate, or was this particular extract poor? Is the hospital interested in sourcing the plants only from a specific region? It could be an explanation for the statistically significant differences between batch 20210201 and 20210502.
    4. The authors did not present the results of the stability studies. Also, I would argue with the conclusion of "high stability." The sample was injected for up to 72 h only. Therefore, we may conclude that it is stable up to this time (if the amount did not decrease by more than 10%). Next, the sentence "the stability of the proposed method." It is not the stability of the method that we are testing but the stability of the sample. It would be best to ask in what conditions CAROL is stored routinely and test if the analytes are stable during the storage time.
    5. How do the authors address matrix effects? The standard curves were prepared in water, while many other compounds present in the extracts can affect the signal. Therefore, the measurements may not be accurate.
    6. Please replace the analyte numbers with the names in the Supplementary Tables for better clarity.
    7. Figure 3: What standard level is presented in B? Adding a chromatogram with LOD or LOQ control could be more informative.
    8. Were the amounts presented in the Supplementary Table within the calibration curve range? In what was the quantity measured? In 1 mL of the extract? In 1 mg of a herb? In the total amount of the herb taken for the extraction?
    9. In Chapter 2.3.1, the authors state that CAROL has higher contents of guanosine and Ganodeic acid A. But higher compared to what?

Reviewer 2 Report

The manuscript is another of those dedicated to establishing the composition of medicines or formulas of traditional Chinese medicine.

There are some aspects that must be corrected or justified in order for the manuscript to be accepted.

Abstract.

Correct Ganoderma acid to ganoderic acid

Introduction:

Figure 1. Check all structures:

Guanosine: One hydroxyl group is missing.

Lucidenic acid: According to the Pubchem database the chirality of the H at position 17 is R

Ganoderic acif F: A methyl radical is missing at position 4.

Ganoderic acid A: The methyl at position 14 must be written in R configuration according to PubChem.

Results.

Figure 2: The MRM response of 2'-deoxyuridine does not appear to be optimal. And it is surprising that the validation parameters in Table 1 reflect the opposite. This incongruence should be justified.

Better organize Table 3 and identify in the chromatograms A, B and C.

Materials and methods.

3.4.1. Spelling CAROL correctly

3.4.3. “For the ethanol extract, the extracted solution was obtained following the same oper-ation as the water extraction method. Then, the absolute ethanol was evaporated in a vac-uum drying oven at 60 °C for 24 h. Afterward, the dry solid was dissolved in 1 mL of methanol, carefully transferred into a 50 ml volumetric flask, and made up to the mark with water.”

I don't understand the reason for preparing an extraction with ethanol and then redissolving in methanol. Is there any additional advantage? Are there interferences soluble in methanol but not in ethanol? It is easier and cheaper to perform the extraction in methanol directly.

All text.

In some cases ganoderic acid is written in lower case and in others in upper case. Please use only the lower case letter.

Reviewer 3 Report

Review on “Study on quality control of Compound Anoectochilus roxburghii (Wall.) Lindl. by liquid chromatography tandem mass spectrometry”

This paper describes a development of a quantitative method to determine eight bioactive compounds in Compound Anoectochilus roxburghii (Wall.) Lindl. oral liquid (CAROL) using LC-MS/MS. The research subject of this manuscript is interesting to readers of Molecules. However, in this manuscript, several severe defects should be addressed. Therefore, the reviewer recommends re-submission of a revised manuscript on Molecules. The revised paper could be acceptable for publication on Molecules, if the authors revise these defects in the manuscript properly. The following major and minor points would be helpful to revise this paper.

Major points

  1. In this study, several chemical structures of analytes have similar functional moieties (purine, glycosylated pyrimidine ring, ribose, and tetracycline ring). Therefore, not only quantitative MRM transitions should be selected carefully without overlapped precursor/product ions, but also additional MRM transitions (namely qualitative MRM transitions) should be utilized to prevent false positive. As shown in Figure 2, some analytes with similar chemical structures were eluted at similar retention time. Nevertheless, it seemed that the authors used only quantitative MRM transitions in this study. The authors should describe additional (qualitative) MRM transitions for this method and further provide on Table 1.

  1. As an IS, tenofovir was analyzed in positive ion mode near RT 3 min. Several analytes detected near 3 min in positive ion mode could be compensated by tenofovir, while other analytes (lucidenic acid A, ganoderic acid A and F) detected after 11 min in negative ion mode cannot be compensated using tenofovir. (Although ganoderic acid F is detected in positive ion mode, elution time of ganoderic acid F is too far from that of IS.) The authors should address this problem.

  1. On Table 3, several quantification results (concentration levels) are much higher than calibration ranges of this method. The authors should describe how calculation contents.

Minor points

  1. On author name list, please check last author.

  1. On overall manuscript, please check reference citation style (superscript).

  1. On Figure 1, the authors used different numbers for analytes, compared to Figure 2 and supplementary Tables. Please revise. Furthermore, on supplementary Tables, please add analytes names.

  1. On third sentence in page 4, the authors did not provide preliminary test results. However, the reviewer thinks that several optimization results about chromatographic separations and MRM determinations are important to describe development of quantitative analytical methods. If possible, please add more data as supplementary information.

  1. On Table 1, the authors provide two places of decimals for precursor/product ions. The reviewer thinks that only one place of decimals would be sufficient.

  1. On fourth sentence in page 5, please revise ’86.42%’ into ’85.07%’.

  1. On Figure 3, please add labels (such as A), B), and C)) on each MRM chromatogram.

  1. On Table 3, the authors described “five batches” on table caption. However, Table 3 provided only four batches. Please revise.

  1. English language needs to be improvement.